# Recent Advances in Tilapia Production for Sustainable Developments in Indian Aquaculture and Its Economic Benefits

Manikandan Arumugam [1], Sudharshini Jayaraman [1], Arun Sridhar [1], Vignesh Venkatasamy [1], Paul B. Brown [2], Zulhisyam Abdul Kari [3,4], Guillermo Tellez-Isaias [5] and Thirumurugan Ramasamy [1,*]

1 Laboratory of Aquabiotics/Nanoscience, Department of Animal Science, School of Life Sciences, Bharathidasan University, Tiruchirappalli 620024, India
2 Department of Forestry and Natural Resources, Purdue University, West Lafayette, IN 47907, USA
3 Department of Agricultural Sciences, Faculty of Agro-Based Industry, Universiti Malaysia Kelantan, Jeli Campus, Jeli 17600, Malaysia
4 Advanced Livestock and Aquaculture Research Group, Faculty of Agro-Based Industry, Universiti Malaysia Kelantan, Jeli Campus, Jeli 17600, Malaysia
5 Center of Excellence for Poultry Science, University of Arkansas, Fayetteville, NC 72701, USA
* Correspondence: ramthiru72@bdu.ac.in

**Abstract:** Tilapia is a widely cultured species native to Africa; these fish are prolific breeders and constitute an economically important fish species supplying higher-quality protein. To meet the global food demand and achieve the UN's Sustainable Developmental Goals (SDG), the aquaculture industry has conceived of productive solutions with the potential for adaptability, palatability, and profitability. Tilapia may play a vital role with respect to the possibility for sustainability in the nutrition and aquaculture sectors. India contributes to the promotion of aquacultural practices through a structural framework focused on agricultural, environmental, geographical, and socio-economic factors that provide opportunities for tilapia farming. Globally, the Indian aquaculture sector is currently the second largest in terms of aquacultural production but is moving toward different species that meet SDG and facilitate international marketing opportunities. The farming of aquacultural species with innovative technology constitutes an efficient use of resources. Productive research on feeding, disease management, construction, and layout helps overcome the challenges faced in aquaculture. These focused and sustained factors of the aquaculture industry offer a latent contribution to global food security. This review reports on the state of the art, the challenges regarding tilapia aquaculture in India, and the Indian government's schemes, missions, subsidies, projects and funding related to tilapia production.

**Keywords:** blue economy; disease management; fish nutrition; species selection; tilapia aquaculture

**Key Contribution:** The present review deals with the important farming strategies of tilapia aquaculture in India. Also, the policies framed by the Indian government through various programs and subsidies to expand the blue economy relating tilapia farming and their direct benefits to the aquaculture farmers were highlighted.

## 1. Introduction

Aquaculture plays a pivotal role in meeting the United Nations' SDG of alleviating poverty (SDG 1) and global hunger, ensuring food security and the provision of adequate nutrition (SDG 2), and promoting sustainable socio-economic growth (SDG 8) [1]. The farming of aquatic organisms in inland and coastal areas improves the local supply of food and the economy. Asia is the leading producer of seafood, producing at a rate of more than 6% per annum [2]. This is due to the increase in the per capita consumption of fish. To meet the SDG and provide food to those in need and economic opportunities in rural areas, culturally appropriate species of fish and production approaches must be

identified. These needs are increasingly being fulfilled by tilapia. Wing-Keong et al. [3] stated that tilapia is one of the most important species of fish in aquaculture, which is capable of filling the gap of the increasing worldwide demand for protein sources. Tilapia farming is widespread, occurring in more than 135 countries and territories [4]. Production is increasing because of tilapia's large size, fast growth, prolific breeding characteristics, palatability, and relatively low cost for production [5]. Although tilapia is a freshwater species, it can tolerate osmotic and alkalinity stresses up to a particular range [6] as well as low dissolved oxygen concentrations and osmotic and alkalinity stress [7]. These fish can mature within 2–3 months of hatching and produce 75–1000 offspring every 22–40 days. Nile tilapia have been cultivated widely in many parts of the world; they are considered one of the first fish species to have been cultured and their cultivation constitutes the largest of the tilapia industries. Globally, Nile tilapia started being cultivated more than 3000 years ago [8]. The Mozambique tilapia industry is the second largest tilapia industry based on its production and exportation rates. The World Bank [9] projected that global tilapia production will reach 7.3 million tons by 2030, an increase from the 4.3 million tons reported in 2010. India's share of global fish production amounted to 5.68% from 2016–2017, corresponding to about 10.79 million tons. Tilapia are preferred over carp because of their firm, white flesh and lack of intermuscular bones. Based on their reproductive behavior, the commercial species of tilapia have been classified into three major categories: (1) maternal mouth brooders (*Oreochromis* species); (2) paternal and biparental mouth brooders (*Sarotherodon* species); and (3) substrate incubators (*Tilapia* species) [10]. The most common commercially farmed species are blue tilapia (*Oreochromis aureus*), Mozambique tilapia (*Oreochromis mossambicus*), Nile tilapia (*Oreochromis niloticus*), longfin tilapia (*Oreochromis macrochir*), redbreast tilapia (*Tilapia rendalii*), redbelly tilapia (*Coptodon zilli*), Sabaki tilapia (*Oreochromis spilurus*), three-spotted tilapia (*Oreochromis andersonii*), and Jaguar guapote (*Parachromis managuensis*). Numerous hybrids have been developed and evaluated, and monosex populations can be developed for various species. The production of various hybrids is also increasing [11]. India's contribution to the yearly annual rate of aquacultural food production amounts to 7.56%, which is greater than the global average from 2000 to 2018 [12]. Thus, this review attempts to study the state of the art and challenges of tilapia culture in India and elaborate the development of the technology that drives this critical food production system in a sustainable manner. The governmental program, Neel Kranti, also known as the blue revolution mission, is a centrally sponsored initiative with the objective of doubling the production and tripling the exportation of fish by 2022. This program began in 2014 and was designed to encourage the use of sustainable and integrated approaches for the development of aquaculture [13]. The main focus of this mission is the utilization and promotion of technological advancement in aquaculture for national food-related and nutritional security. The ultimate goal of the program is to encourage the use of sustainable and integrated approaches for the development of the fisheries sector in India. [14]. This initiative has four major components: strengthening infrastructure and security at ports and harbors, boosting skill development and training for fishermen, encouraging aquaculture, and ensuring fishermen have greater access to financial facilities. The program's infrastructure enhancement component intends to offer better facilities at ports and harbors, mobile health services, and a fishing insurance plan. This will allow fishermen to carry out their activities in a safe and secure manner and enhance their overall living conditions. The associated training program seeks to educate fishermen with respect to the optimal practices in water safety, fishing equipment maintenance, and current fishing techniques. To support aquaculture, the initiative will allow fishermen to receive improved technical support, thereby allowing them to launch their own fish-farming companies. This program will assist fishermen in purchasing higher-quality seeds and gaining access to more markets, thereby improving their revenue and providing additional job prospects in the industry. In summary, the Indian government's Neel Kranti program is a much-needed effort that can dramatically increase the country's aquacultural output and assist fishermen in securing superior economic and living conditions [15–17]. Though India is the largest

producer of Indian carp (*Catla catla, Labeo rohita,* and *Cirrhinus mrigala*), the global demand and consumption of tilapia have paved the way for the enrichment of the productivity of Indian aquaculture [18]. In this regard, the government of India has set forth detailed agro- and socio-economic guidelines for the cultivation of this non-native species with the goal of protecting native inland species and their production. The guidelines encourage the amassment of collective knowledge and the undertaking of interdisciplinary efforts, including mathematical modelling systems, Internet-of-things (IoT)-based in-silico approaches, geospatial technology, fisheries and engineering technology, and management strategies, to provide an innovative and productive outcome regarding the production of tilapia from the aquaculture industry. The government of India also provides subsidies and development funds to facilitate tilapia farming based on the poverty line for farmers. In this review, specific facets such as tilapia aquaculture, the contribution of the Indian tilapia industry to global aquaculture, major production guidelines, various culturing methods, species-specific selection criteria, feed and disease management strategies, and the development of projects/schemes for tilapia production in India will be discussed.

## 2. Tilapia Aquaculture in India

Nile tilapia is the primary cultivable species in India. This cichlid was initially introduced in the state of Kerala, while Mozambique tilapia were imported in 1952 and stocked into reservoirs and ponds in Kerala state [19]. Due to their rapid rates of reproduction, the fish overpopulated the area and slowly migrated into the reservoirs of Tamil Nadu, Karnataka, and Rajasthan, resulting in the extinction of certain inland fish species, such as *Tor tor* and *Tor putitora*. In 2005, the Yamuna River harbored a certain quantity of Nile tilapia; due to this species' characteristic reproductive behavior, the abundance of tilapia increased in comparison with the total fish species in the river by 3.5% in 2 years (reported by the National Fisheries Developmental Board) [20]. Johnson et al. [21] reported that a drastic increase in the catch percentage of tilapia ranged from 6.7% to 85.9% from 2008 to 2018, which is expected to reach >90% according to their decadal species composition study. The experimental study also noted the species diversity of Nile tilapia from the total catch in the Halali reservoir [22]. The introduction of tilapia via a polyculture strategy also reduced the average weight of other major carp. Panikkar [23] recommended the formulation of a national policy, which led to a ban on tilapia propagation. The strict guidelines on tilapia farming in India have resulted in a renewed interest in the cultivation of several species, including *Oreochromis mossambicus*, *Oreochromis niloticus*, *Oreochromis urolepis*, and *Captodon zillii*, which are now available throughout the country [24]. Globalization, the food demand within India, and economic development opportunities precipitated the current situation, which, consequently, facilitated tilapia farming under the guidelines discussed below. The relevant regulatory entities in this regard include the Department of Fisheries, the Central Institutes for Marine and Inland Fisheries Research, the Rajiv Gandhi Centre for Aquaculture (RGCA), the National Fisheries Development Board, and other government agencies. Thus, tilapia is now farmed with sustainable farming technology by following the respective government-issued guidelines.

### 2.1. India's Contributions to Tilapia Production

The State of World Fisheries and Aquaculture has acknowledged the stupendous growth of the Indian fisheries sector, as it ranks, globally, fourth in terms of capture fisheries and second in terms of inland capture fisheries, contributing as much as 14% of the share of the total global inland capture [25]. The Indian government has launched a number of initiatives and projects to boost aquacultural output in the country. The Blue Revolution Plan, the National Fisheries Development Board (NFDB), and the Fish Farmers Development Agency (FFDA) are among the major projects. Reflecting and driving the global shift from capture to culture, the report underscores the fact that 57% of India's total fish production stems from aquaculture. The inland and marine sectors provide a wide range of water resources for culture and capture fisheries. In 1950–1951, India's total fish

production was 0.75 million metric tons (MMT); then, it drastically increased to 9.5 MMT in 2012–2013. Moreover, the current production level has reached 16.25 MMT due to the projects and schemes funded by the Indian government [25]. The aim of the Blue Revolution Program is to boost fish output by building fish farms, hatcheries, and processing facilities. The National Fisheries Development Board promotes sustainable aquaculture methods and assists relevant businesses financially. The Fish Farmers Development Agency seeks to boost the productivity of fish farmers by offering training and assistance. These measures have resulted in tremendous development in the aquaculture sector, increasing employment and strengthening the country's export revenues [13]. The GOI aims to double the income of fishers, fish farmers, and fish workers over five years, with a 9% annual growth rate, to attain the fish production target of 22 million tons by 2025. This scheme, with reservoir fisheries as one of the focus areas, aims to create additional employment opportunities, both directly and indirectly, for six million people employed in the fisheries industry and its allied activities [26]. The Food and Agriculture Organization (FAO) has predicted that India's fish production level will grow by 26% between 2018 and 2030, which is 6.8% and 11.5% faster than the projected growth rate for Asia and the world, respectively [27].

*2.2. Guidelines for Tilapia Culture in India*

In aquaculture, efforts to increase the productivity of tilapia resulted in high population density, which, in turn, caused outbreaks of Tilapia Lake Virus (TLV). Although tilapia farming has resulted in adverse environmental impacts on native fish species, tilapia have also become a prominent species whose consumption allows rural communities to meet their food and nutritional requirements. Thus, the National Committee approved the introduction of exotic aquatic species such as Nile tilapia in 2006. However, farmer-friendly guidelines for tilapia were not implemented until December 2011. These guidelines were established based on the concept of the monitoring (M), control (C), and surveillance (S) of the hatchery, nursery, and farming practices of tilapia culture in India [28]. The detailed guidelines for farming tilapia in India can be found on the Department of Fisheries website maintained by the Ministry of Fisheries, Animal Husbandry, and Dairying, Government of India (www.dahd.nic.in, accessed on 20 January 2023). A Steering Committee was established at the department of fisheries at the national level to monitor tilapia seed and grow-out production. The guidelines of the committee initially dealt with cage farming, which, subsequently, requires registration and information on location, the area of culture, the type of culture and its intensity, the size of the seed to be stocked, the stocking density, and the biosecurity parameters in both cage-based and intensive culture. For subsidies and governmental funds, the guidelines should be followed strictly, with particular emphasis on stocking density and biosecurity.

**3. Farming Strategies of Tilapia Culture: The State of The Art**

The use of appropriate and proven farming strategies for tilapia aquaculture facilitates better yields and utilization of resources [29]. Several technological advancements are widely used in aquaculture to overcome various challenging factors, such as climate change, land availability, socio-economic concerns, and environmental barriers. Various studies have been reported concerning the strategies and efficient practices for the successful production of tilapia. These practices include Biofloc technology (BFT); backyard brackish water aquaculture; recirculatory aquaculture systems (RAS); cage culture systems for the farming of potential high-yield varieties of tilapia such as the Genetically Improved Farmed Tilapia (GIFT) strain, hybrids, and monosex populations; and Integrated Multi Tropic Aquaculture (IMTA). Polyculture (multiple species in the same production system) and integrated fish farming (fish farms integrated with terrestrial agricultural crops) provide additional income to farmers. One study reported that the integration of aquaponics with BFT applied to GIFT tilapia and bell peppers resulted in improved production without affecting growth or stress parameters [30]. This technological advancement helps overcome the challenges in the agro-aquaculture sector [31]. The integration of BFT and

RAS resulted in better resource utilization and production by providing supplementary feed for Nile tilapia [32]. Oparinde [33] developed a mathematical model to address the adaptation strategies associated with changes in the climatic conditions for aquaculture. Geographical-Information-System (GIS)- and remote-sensing-technology-based data are associated with applications for effective farming, land or site suitability assessment, or resource availability. The GIS-based (AHP-Analytical Hierarchical Process) approach facilitates geospatial mapping for the planning or construction of fish farms and the use of brackish water resources [34]. Hence, the application of these technological advancements in aquaculture paves the way for sustainable farming practices. The following strategies concern the improved farming practices applied in tilapia production.

### 3.1. Recirculatory Aquaculture System (RAS)

An RAS uses biofiltration to eliminate trash and raise oxygen levels, thus allowing for an extremely efficient and eco-friendly approach. Initially, cleansed water is treated with chemicals to remove chlorine and other hazardous compounds [35]. After this water has been treated, it then passes into the fish tanks or raceways where the fish are cultured. These tanks are often constructed with space to swim while also allowing for effective water flow through the system. The water becomes tainted with ammonia, nitrites, and nitrates as the fish create waste; this waste can be passed through a biological filter, which is a series of tanks containing beneficial microorganisms, to decrease such impurities. These bacteria convert ammonia and nitrites into nitrates, which may be utilized as fertilizer for plants. The major aerobic bacteria involved in this system belong to the genera Nitrosomonas, Nitrosococcus, Nitrosospira, or Nitrosolobus. These bacteria tend to convert nitrite to nitrate (Figure 1). The water is constantly pumped through the biological filter and back into the fish tanks, thereby ensuring that the fish have a healthy aquarium habitat. The mechanism of the recirculation system reduces water usage significantly, rendering it a more sustainable approach than standard aquacultural methods. To maintain ideal water quality, some RASs contain additional water treatment procedures, such as protein skimming, carbon dioxide level monitoring, or UV sterilization, in addition to biofiltration, thereby increasing the potency of water quality maintenance. In summary, an RAS is a highly efficient and environmentally friendly method for raising aquatic plants and animals [36–38].

**Figure 1.** Reaction mechanism of ammonia−nitrite oxidation used in Recirculatory Aquaculture System (RAS).

RASs are closed systems that conserve water by recycling and are capable of affording super-intensive production levels (Figure 2). One of the plausible solutions to the water crisis and problems regarding land utilization in urban areas is RAS technology. Ye et al. [39] developed a statistically based imaging technique for tilapia farming in an RAS. Shnel et al. [40] designed the zero-discharge RAS production system for tilapia. In this method, nitrogen removal was performed by a fluidized bed reactor. A rotating biological contactor device for tilapia was used to manage water quality and remove ammonia in

an RAS production system [41]. An RAS provides optimum environmental conditions year-round and may be one of the best solutions for the climatic crisis currently threatening aquaculture [42]. The production of holy basil (*Ocimum tenuiflorum*) and Nile tilapia resulted in a better growth rate of tilapia and an improved holy basil yield [43].

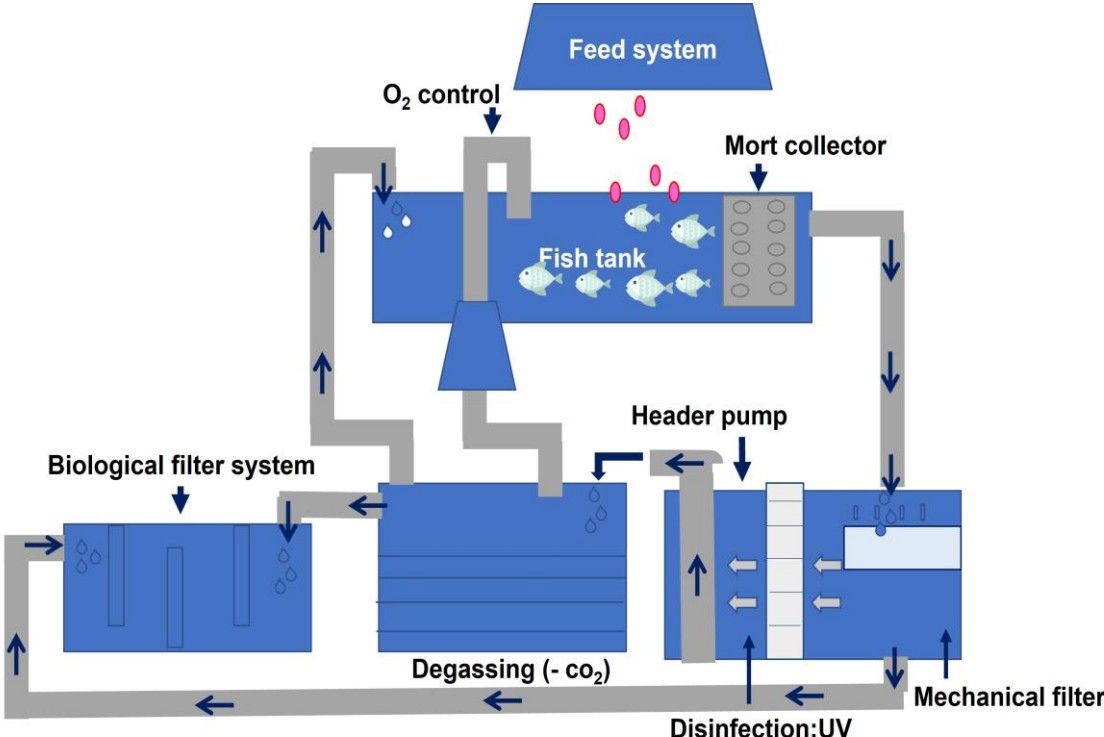

**Figure 2.** Typical representation of tilapia−oriented Recirculatory Aquaculture System (RAS).

*3.2. Biofloc Technology (BFT)*

BFT is also known as the activated suspension technique (AST); it involves the use of microbial communities to break down waste particles and transform them into a protein-rich biomass that can be easily consumed by the fish [44]. The process is reliant on the production of high levels of organic matter that results in high concentrations of suspended solids. However, these solids provide a surface for bacterial colonization, and these bacteria then serve as a food source for the fish [45]. By utilizing this biofloc technology, farmers can create a self-sustaining system that increases the efficiency of production and reduces their dependency on external inputs, thereby reducing their overall operational costs. The technology has been shown to effectively improve yields, reduce costs, and ensure the sustainable production of tilapia [46,47] (Figure 3).

This process serves as a source of food for fish [48]. The addition of carbon (C) and nitrogen (N) sources and the constant aeration and agitation of the water column result in the superior production of natural feed for the cultured aquatic species. The optimum ratio of C to N in BFT is 10:1 [49]. The use of BFT helps reduce the environmental impacts of aquaculture. The formulated diets and their ingredients can constitute an effective and sustainable farming technique for producing commercially valuable species in aquaculture [50]. Effluents from BFT can also be used in an aquaponic-based system ("flocponics"); this feasible approach enhances the growth of tilapia more than that of the plants [51]. The use of BFT has been shown to improve the quality of larvae and brood fish [52].

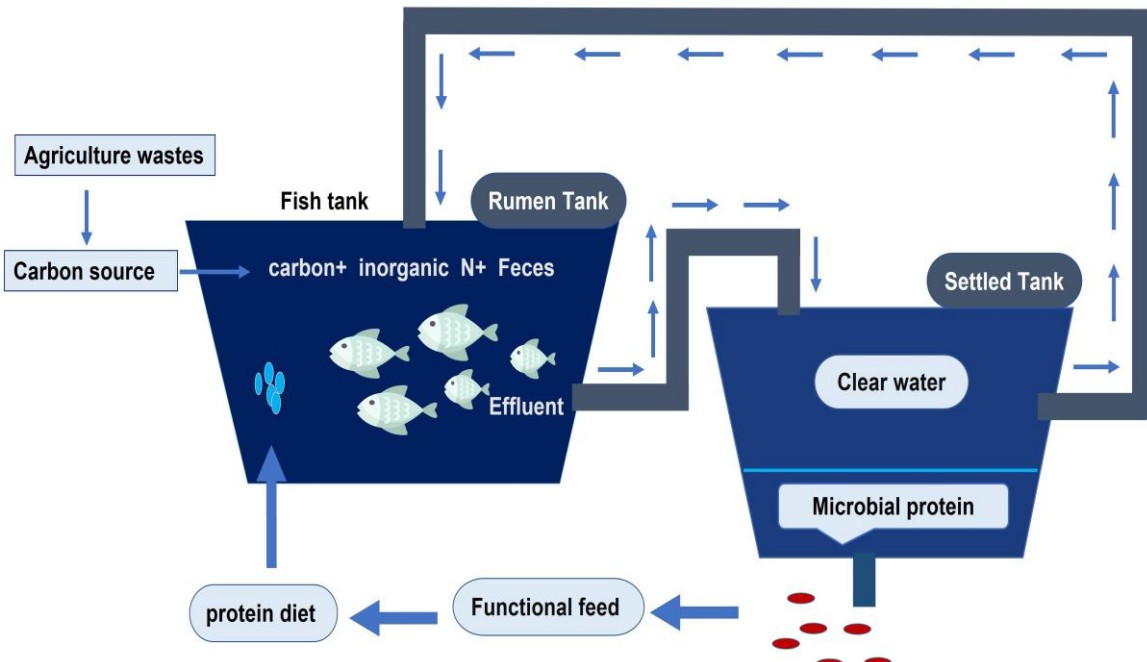

**Figure 3.** Typical illustration of tilapia farming using Biofloc Technology.

Based on a partial production cost analysis and a subsequent investigation, Luo et al. [53] reported that the production of GIFT tilapia was more lucrative when using BFT compared to a traditional RAS. A zero-water exchange system using BFT resulted in optimum growth and hematological and immune parameters in GIFT strain fish [54]. The indoor tank cultivation of Nile tilapia using BFT resulted in 100% survivability and an increased production rate [55]. Certain studies have suggested that the biofloc system reduces the entry of pathogens due to the recycling of nutrients and water [56,57] and that the flocs produced by this technology can enhance the amount of protein available for the tilapia to consume, leading to a reduction in the usage of feed [58]. This type of approach reduces the costs of production and generates greater profits [59].

*3.3. Cage Farming*

Cage culture in open water is another production system that is particularly well suited for the introduction of aquaculture in rural areas or for adoption by farmers with little aquaculture experience [60]. The major advantage of cage culture is that it can be implemented in existing water bodies such as rivers, lakes, ponds, seawater, etc. In addition, it provides an excellent environmental sustainability index, allowing for affordances such as lower usage of resources and reduced pollutant accumulation [61]. Formulated feed is commonly fed to fish housed in cages. In cage culture, fish require significant feed supplements, including formulated feed, to promote growth, health, and productivity. The GIFT strain is productive in cage culture systems. The use of sterile, monosex male tilapia (*Oreochromis niloticus*) is permitted in cage culture in India [62]. The farming of tilapia in ponds and cage culture is prominent and gaining popularity in India, wherein the focus is on Nile tilapia [57]. Seed, larval, and brood quality and stocking density play essential roles in the success of tilapia cage culture. Stocking density is vital for production, disease, and stress management in a fish culture environment. This intensive culture method has certain guidelines in the Indian regime for the culture of monosex tilapia, GIFT, hybridized, and hormonal sex-reversed tilapia, which have been designed to impede the prolific breeding tendency of the tilapia. Chakraborty et al. [63] evaluated the stocking density and growth of Nile tilapia in the Gangetic plains, India. They recommended a stocking density of kg/m$^3$ for caged-cultured mono-sex Nile tilapia in the Indian context. Another important factor in tilapia cage culture is feed management. Feeding and nutrient management in

cage culture involves artificial and natural feed. Providing natural feed (phytoplankton and zooplankton) improves the nutritional quality of the fish and reduces the necessity for the supplementation of artificial feed in cage culture. Periphyton is a natural food source that is gaining popularity in cage culture as it reduces the protein requirements necessitated by commercial feed and functions as a complementary feed for the fish [64]. According to Delphino et al. [65], streptococcus-resistant tilapia cultured in cages were found to present ≤ 10% mortality, which significantly increases the production rate by preventing a streptococcus infection. The major drawbacks of cage culture are environmental impacts such as the release of nitrogen, nutrients, and pollutants in waterbodies by uneaten feed [66].

*3.4. Polyculture Tilapia Farming*

The culturing of more than one species of aquatic organism in the same system is called polyculture. This approach facilitates the better utilization of the available natural feed in ponds by using species displaying different food habits (foraging), thereby facilitating higher fish production per unit area [67,68]. Polyculture systems can also be referred to as co-culture, multi-trophic, or integrated aquaculture farming systems. However, the systemic approaches differ in each system [69]. The primary and secondary species in a polyculture system enable cost-effective production [70]. Tilapia has a shorter growing period (a maximum of 6 months to reach 500 g in body weight) when compared to other teleost species; thus, the cultivation of tilapia with other species requires specific techniques and strategies {68]. Detailed guidelines and recommendations regarding the species cultured with tilapia polyculture facilitate better income without affecting species production. Tilapia have been successfully co-cultured with crustaceans (prawn/shrimp) and other teleost fishes such as silver carp (*Hypophthalmichthys molitrix*) and common carp (*Cyprinus carpio*) [71]. When tilapia are co-cultured with shrimp/prawns, the tilapia are able to act as an effective filter feeder by consuming zooplankton, while the leftover phytoplankton are consumed by the shrimps/prawns, thereby reducing the formation of algal blooms and enhancing economic value [72]. Hisano et al. [73] reported that the co-culturing of Nile tilapia and giant prawns (*Macrobrachium rosenbergii*) in a BFT-based RAS polyculture system resulted in better feed and protein utilization for the tilapia. However, in a polyculture system consisting of a combination of tilapia and carp, the tilapia achieved greater growth than the carp due to the reduced feed conversion ratio [74]. Similar results were obtained by Papoutsoglou et al. [75], where the ratio of 40:60 carp/tilapia production resulted in better growth with a lower FCR (Feed Conversion Ratio) and carcass lipid concentration. In fertilized ponds, the mortality rate of tilapia was higher than that of carp [76]. Previous studies have suggested that management approaches incorporating parameters such as stocking densities, species, the age of the species, and feed and niche requirements are essential in the polyculture farming of tilapia [68,77–79].

*3.5. The Integrated Farming of Tilapia*

Integrated fish farming involves the combination of farming fish with livestock or other terrestrial agricultural animals. In this approach, the systems are linked to each other; thus, land and water resources are efficiently used, and financial and labor costs are reduced. Integrated fish farming commonly incorporates waste or by-products from the terrestrial side for utilization on the aquatic side. The overall outcome of the integrated farming system is a high yield with low input and a limited amount of supplementary feed required for the fish [80]. Zoonotic pathogen sources and organic manure can contaminate soil and water in an IFS (environment) with dangerous chemicals and pathogens that pose a threat to human health [81]. Concerns regarding environmental risks and the bioconcentration of harmful substances should be mitigated to achieve sustainable IFSs [82]. Adverse effects on an IFS should be reduced by adopting and adapting environmentally friendly approaches that are eventually safe and hygienic and prevent further environmental degradation [83].

*3.6. Integrated Multi Tropic Aquaculture (IMTA)*

An expansion of the Integrated Farming System has been developed and termed integrated multi tropic aquaculture. IMTA is commonly practiced as a semi-intensive culture method that is widely used for the cultivation of animals feeding on diverse trophic grades (Figure 4). Waste nutrients are collected as sediments in this system and are utilized by other organisms. This strategy involves the use of filter feeders to remove excess feed to avoid environmental water pollution [66]. In IMTA, species from different niches consume the available resources; hence, the nutrient inputs become more efficient [72]. IMTA practices are of several kinds and have also been called Integrated Peri-Urban Aquaculture Systems (IPUASs), Integrated Agriculture Aquaculture Systems (IAASs), and Integrated Fisheries and Agriculture Systems (IFASs) [84,85]. David et al. [86] reported the results of the cultivation of Amazon River prawn (*Macrobrachium amazonicum)* and Nile tilapia using the IMTA technique. In this study, the Nile tilapia acted as a feeding organism, whereas the Amazon River prawn acted as a recycler. It has been reported that IMTA could be used as an environmental stability agent in the Sundarbans, serving as a balance between food production while also supporting the ecological security of the mangrove ecosystem [87]. When applied to floating cage systems, IMTA approaches enhance the growth and production of tilapia [88]. According to Rodrigues et al. [89], integrated farming incorporating tilapia and the Amazon River prawn results in higher growth rates when natural live feed is utilized. In an integrated farming strategy, the size of the species plays a vital role. When prawns and tilapia are cultured via integrated farming, the size of the prawn will increase due to the increased uptake of phytoplankton [90].

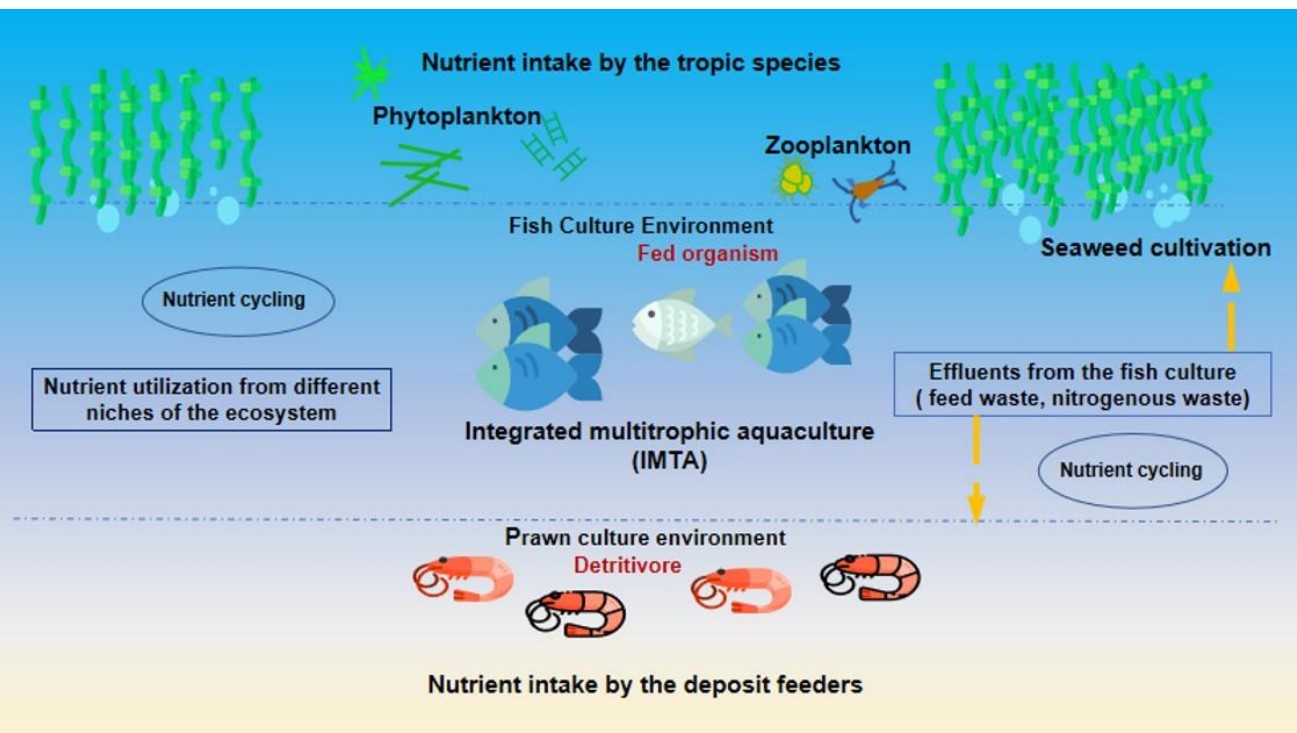

**Figure 4.** Graphical illustration of IMTA depicting efficient utilization of nutrient recycling system.

## 4. Strategies for Species Selection in Tilapia Farming

*4.1. Farming of Monosex Tilapia*

When undertaking the farming practices of tilapia culture, farmers face unrestrained reproduction. To overcome this limitation, monosex tilapia cultivation has been implemented [91]. The monosex production of tilapia is a rapidly growing and popular technique in the field of tilapia farming (Figure 5). This technique is widely used due to the uniform size of these fish, which are also gaining popularity among consumers [63]. The novel

production and masculinization of Nile tilapia involves crossing the YY male genotype with XX females (wild). This technique is known as genetically male tilapia or YY male tilapia technology. This method is also used as a male factorial sex-determining mechanism [92]. Other methods involved in monosex production include hormonal sex reversal, interspecific hybridization, and the production of supermales and genetically improved varieties [93]. Male tilapias grow more quickly than females and use less metabolic energy to obtain a uniformly sized output; hence, these practices lead to the production of males at a higher rate for monosex populations [94]. Androgenesis, triploidy, and transgenesis methods are also available [93]. These methods have the potential to transform tilapia production by allowing farmers to produce males or females based on their preferences, removing the need to sort and eliminate fish [95]. They also have the potential to provide considerable economic advantages to farmers while contributing to the expansion of the blue economy [96]. However, the adoption of these methods raises queries about food safety, environmental effects, and ethical problems. As a result, adequate laws and standards must be implemented to reduce possible hazards related to the usage of these approaches [97]. The proper usage of these strategies may aid the expansion of the blue economy and bring economic advantages to farmers while also ensuring the industry's safety and sustainability [96]. The monosex production of tilapia is an ongoing line of research. The study conducted by Sayed and Moneeb [98] indicated that the nonsteroidal aromatase inhibitor Fadrozole could be used to produce male populations of fish. The synthetic male hormone 17α-methyltestosterone is used to reverse the sex of tilapia and produce monosex populations. Considering the negative health-related effects of using synthetic hormones for sex reversal, it has been recommended that they be substituted with pyotosterols [99]. Ghosal et al. [100] suggested that the ethanolic extract of *Basella alba* leaf and the methanolic extract of *Asparagus racemosus* can be used as safe and eco-friendly alternatives for synthetic sex reversal hormones for monosex Nile tilapia.

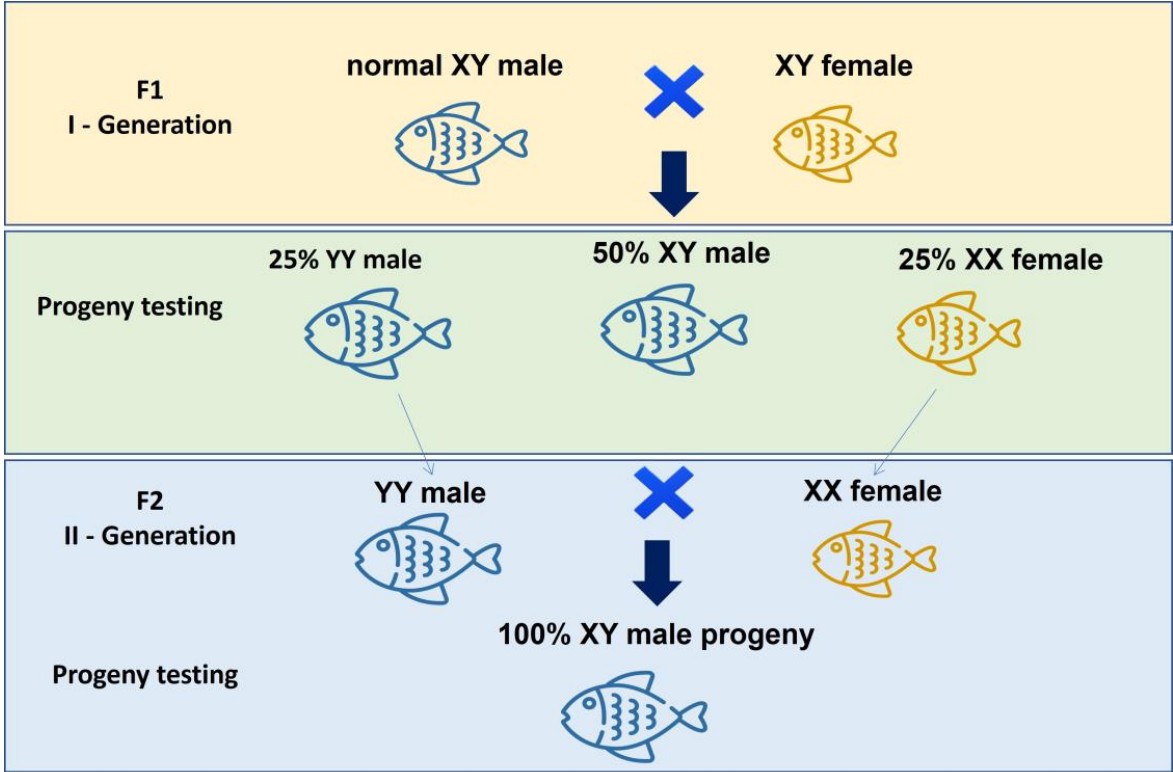

**Figure 5.** Male tilapia (monosex) production through YY male tilapia technology.

### 4.2. Farming of GIFT Tilapia

GIFT tilapia were successfully developed by the International Center for Living Aquatic Resources Management (ICLARM), which is now known as the World Fish Centre (WFC), and its allies [101]. In India, the GIFT strain is an improvement compared to other available strains. The main aim of introducing the GIFT strain is to achieve high yields, rapid growth, and high rates of survival at a low cost. The value offered by monosex GIFT strains of tilapia resulted in wide-ranging adoption in Asian countries [102]. It has been reported that the improved variety of Nile tilapia tolerates both saline and freshwater without affecting the FCR, growth, or gill conditions of the fish [103]. One study suggests that an around 27–36% faster growth rate can be achieved in GIFT tilapia compared to non-GIFT tilapia by using mono and polyculture strategies [104]. The detection and identification of potential genes for the improvement of the cultured organism offers significant potential for further improvement. Thus, genetically based selective breeding with the aid of genome sequencing and mapping will pave the way toward the improved production of GIFT tilapia. Xia et al. [105] constructed selection footprints and a genome-wide map of genetic variation in tilapia. These tools could be used to help construct new and productive varieties of genetically improved tilapia by using markers such as DNA markers, thereby constituting a significant contribution to the production of fish species.

### 4.3. Farming of Hybrid Tilapia

Two of the more common hybrid tilapia are the red tilapia or golden tilapia. The *Oreochromis mossambicus* × *Oreochromis niloticus* hybrid is gaining popularity because of ease with which its cultivation can be managed [106]. Beardmore et al. [93] indicated that hybridization can result in monosex populations. Based on the performance of analyzed the fish, they found the gene and regulatory pathways related to osmoregulatory tolerance in hybrid tilapia [107]. The advancement of the production of hybrid tilapia requires one to understand the genetic linkages of the parental strains (the ancestry) of these fish [108]. Gene-sequenced omics and computational investigations aid the development of productive hybrid tilapia strains. [109]. Avallone et al. [110] developed an simple and inexpensive method called Local Ancestry Inference (LAI) for a tilapia-breeding program using Digest-RAD-sequence-derid Single Nucleotide Polymorphism Markers (SNPM). The goal of their experiment was to trace ancestral genes via a fast and accurate method for the production of potential high-yield and disease-resistant varieties of hybrid tilapia. This method helps remove the unwanted traits in fish [111]. The selective breeding of tilapia to produce hybrid varieties leads to the optimal presentation of economically and environmentally favorable traits [112].

### 5. Management of Feed and Nutrients

To maintain optimum growth and immune functions, feed should contain energy and nutrients that meet the requirements for tilapia culture [113]. Nutrients play a vital role in the regulation of metabolism and the maintenance of homeostasis in fish [114]. Various parameters, such as body weight increase (BWI), FCR, the protein efficiency ratio (PER), specific growth rate (SGR), and weight gain (WG), are used to measure growth as a function of feed offered. Fishmeal is a major source of nutrients in fish feed. However, due to the depletion of fishmeal stock and fluctuations in its selling price, investigations are already underway to find a suitable alternative. Mostly plant-based alternatives are preferred because of their nutritional profile and abundance. Nevertheless, the antinutritional factors present in plant sources hinder the process of completely replacing fishmeal in fish feed. The dietary needs of tilapia vary based on the developmental stage, water temperature, and fish size [115]. It is critical to balance the diet with the proper macronutrients and micronutrients while avoiding overfeeding, which can cause water quality concerns such as increased fish waste and uneaten food [116]. Producers must also consider feed costs and devise feeding systems that improve economic efficiency while preserving fish growth and quality [117]. Fish feed production is extremely difficult since it frequently necessitates the

exploitation of arable land to grow crops that are then transformed into fish feed [118]. This is a sizeable issue since arable land may be better employed for human food production rather than for fish feed manufacturing. The growing need for fish feed is consuming a large quantity of arable land that could otherwise be exploited to produce food for human consumption [119]. The land used to manufacture fish feed could be utilized to grow crops that could feed humans in many parts of the world, particularly in areas where food insecurity is already a serious concern [120]. As a result, it is critical to investigate sustainable alternatives for the production of fish feed, such as the utilization of insect-based protein sources [121]. This would reduce the strain on arable land and water resources while also providing a long-term source of protein for tilapia production. Researchers have conducted various studies to find plant-based alternatives to fishmeal. Nevertheless, due to antinutritional factors, plant-based alternatives have only been used to partially replace fish meal [122].

**Table 1.** Various feed supplements and their performance with respect to fish health.

| S. No | Feed Supplement | Performance | Fish Species | References |
|---|---|---|---|---|
| 1. | *Tridax procumbens* | Improves growth, production of antioxidants, immunity, and resistance to monogenean parasitic infection | (*Oreochromis niloticus*) Nile tilapia | [123] |
| 2. | Caraway seed | Improves growth performance | (*Oreochromis niloticus*) Nile tilapia | [124] |
| 3. | *Silybum marianum* | Promotes growth and enhances serum biochemical indices, antioxidant status, and gene expression | *Oreochromis niloticus* Nile tilapia | [125] |
| 4. | *Trigonella foenum-graecum* | Improves oxidative status and immune gene expression and histopathology | (*Oreochromis niloticus*) Nile tilapia | [126] |
| 5. | *Salvadora persica* | Improves hematoimmunological parameters and enhances antioxidant responses against *A. hydrophila* infection | (*Oreochromis niloticus*) Nile tilapia | [127] |
| 6. | *Yucca schidigera* | Improves growth performance, hepato-renal function, and antioxidative status and effects histopathological alterations against hypoxia | (*Oreochromis niloticus*) Nile tilapia | [128] |
| 7. | Menthol essential oil | Improves growth performance, digestive enzyme activity, immune-related genes, resistance against acute ammonia exposure | (*Oreochromis niloticus*) Nile tilapia | [129] |
| 8. | Dietary coenzyme Q10 and Vitamin C | Enhances growth, digestive enzyme activity, immune-related genes, and resistance against acute ammonia exposure | (*Oreochromis niloticus*) Nile tilapia | [130] |
| 9. | Soybean meal diet combined with bokashi leachate | Improves feed intake and growth performance | (*Oreochromis mossambicus × Oreochromis niloticus*) Red tilapia | [131] |
| 10. | Enzymatic feather meal | Enhances growth, nutrient retention, and digestibility | (*Oreochromis niloticus × Oreochromis aureus*) | [132] |
| 11. | Organic acid salt blend and protease complex combination | Improves growth and nutrient digestibility | *Oreochromis niloticus × Oreochromis aureus* | [133] |
| 12. | Methylated soy protein isolates | Acts as good immune-modulating substance and improved gut health | (*Oreochromis niloticus*) Nile tilapia | [134] |

| S. No | Feed Supplement | Performance | Fish Species | References |
|---|---|---|---|---|
| 13. | Whey Protein Concentrate (WPC) | Improves gut health, total weight gain, survival rate, and immune status of fish against *Aeromonas hydrophila* | (*Oreochromis niloticus*) Nile tilapia | [135] |
| 14. | *Bacillus subtills* and *Lactobacillus plantarun* | Increases amylase (enzymatic) activity, modulates intestinal microbiota profile | (*Oreochromis niloticus*) Nile tilapia | [136] |
| 15. | *Bacillus pumilus* and exogenous protease | Enhances growth, immunity, serum parameters, gene expression and gut bacteria | (*Oreochromis niloticus*) Nile tilapia | [137] |
| 16. | *Enterococcus faecium* | Improves growth, hematological and biochemical parameters, and non-specific immune response | (*Oreochromis niloticus*) Nile tilapia | [138] |
| 17. | *Aspergillus oryzae* | Improves oxidative status and immune response against hypoxia | (*Oreochromis niloticus*) Nile tilapia | [139] |
| 18. | *Clostridium butyricum* | Improves growth, feed utilization, and gut health | *Oreochromis niloticus* × *Oreochromis aureus* | [140] |
| 19. | Chitosan and chitosan nanoparticles | Improves health and phagocytic activity | (*Oreochromis niloticus*) Nile tilapia | [141] |
| 20. | Zinc oxide nanoparticles | Improves health | (*Oreochromis niloticus*) Nile tilapia | [142] |
| 21. | Dietary sodium butyrate nanoparticles | Enhances growth | (*Oreochromis niloticus*) Nile tilapia | [143] |
| 22. | Synergized selenium and zinc oxide nanoparticles | Improves growth, hemato-biochemical profile, and immune status and reduces oxidative stress | (*Oreochromis niloticus*) Nile tilapia | [144] |
| 23. | Cinnamon nanoparticles | Enhances antioxidant and digestive enzyme activity, growth, and health | (*Oreochromis niloticus*) Nile tilapia | [145] |

Natural organisms, supplementary feed, and feed additives are widely used in commercial fish farming [146]. Depending on the culturing practices employed and the foraging behavior of the specific group, tilapia will grow rapidly when fed with fishmeal-based diets, plant-based diets, biofortified feed additives, or other natural types of feed. The use of formulated diets helps curtail unwanted chemical inputs, and the use of synthetic antibiotics naturally fosters the growth and immune status of the fish [147]. Fish meal is an excellent protein ingredient in diets but is very expensive [148]. Fish meal provides protein and essential amino acids but can also contain thiaminase, an anti-nutritional factor that can degrade thiamine [132]. The demand for fishmeal exceeds the supply and alternative protein sources are needed. Tilapia present positive results when fed with alternative protein ingredients (Table 1). Thus, feasible, balanced, low-cost, anti-nutritional-agent-free feed should be formulated for sustainable aquacultural production. Studies concerning feed formulation and nutrition technology are increasingly relying on proteomics, transcriptomics, genomics, and metabolomics to interpret the efficiency of growth- and immune-enhancing feed formulations in aquatic feed and nutrition [149].

## 6. Strategies for Diseases Management of Tilapia

Disease outbreaks can cause severe losses in aquaculture. Proper diagnostic advancements should be implemented to avert economic loss [150]. Several diseases are caused by poor water quality management, the high stocking of fish, and improper feeding strategies [151]. The continuous usage of antibiotics/medications leads to an increased incidence of drug-resistant bacteria; another consequent drawback is an accumulation of antibiotics in fish [152,153]. Tilapia are highly susceptible to pathogens such as bacteria, fungi, viruses, and ecto- or endoparasites or their secondary toxic metabolites. Tilapia are also highly susceptible to Motile Aeromonas Septicemia (MAS), columnaris, edwardsiellosis, francisellosis, streptococcosis, and vibriosis [5]. TLV (an ortho myxo-like virus) is a potential threat to farming and production [154], and it is ascribable to certain bacterial pathogens such as *Aeromonas, Flavobacterium*, and *Streptococcus*. Certain co-existence studies analyzed TLV and bacterial pathogens to assess the resultant epidemic disease [155]. Ectoparasites that affect tilapia farming include monogeneans (*Cichlidogyrus, Cyrodactylus* etc.,) and protozoans (*Trichodina, Vorticella*), which can result in severe monetary losses in the tilapia industry [156]. These disease-causing agents effect high mortality rates and are a menace to future production [157]. A disease outbreak in tilapia production causes adverse effects on aquaculture (Figure 6). *Streptococcus agalactiae* and *Streptococcus iniae* are the major causative agents for the endemic disease streptococcosis. This disease causes severe mortality, specifically during the summer months when the increase in water temperature favors the growth of *S. iniae* [158]. Ismail et al. [159] reported that vaccine-based diets reduce the severity of streptococcosis infection by as much as 13% and increase survival rates by up to 75%. Biocontainment measures include the quarantining of the diseased fishes, water treatment using ultraviolet light, and chemical treatment (disinfectants) to reduce the risk of diseases in the culture environment before the administration of medication. However, antibiotics, chemical agents, or chemotherapeutics are only used after the identification of sick fish [160,161]. Vaccination and improved hygiene protocols are critical to avoiding antibiotic abuse in tilapia production. Antibiotic resistance occurs when bacteria develop resistance to the actions of antibiotics, rendering them more difficult to treat. Antibiotic overuse in the fish farm industry can result in the entrance of antibiotics into the food chain, thereby potentially compromising human health. Vaccination protects farmed tilapia against infections that might harm them, thus lowering the need for antibiotics. Improved hygiene protocols can also help avert disease outbreaks by lowering the likelihood of pathogen transmission. The implementation of these strategies is assured to promote sustainable and safe tilapia farming both in terms of the environment and human health.

The sustainability of aquaculture requires the control of diseases. The government, NGOs, and various research institutes in India are focusing on this challenge and providing disease-resistant strains of fish [119]. The emerging techniques, such as the sequencing of whole genomes, provide new insights into the disease resistance of high-yield varieties of tilapia. *Oreochromis spilurus* cultured in seawater contains an antimicrobial peptide [162].

### 6.1. Vaccines

Fish are cold-blooded animals but respond to vaccines like warm-blooded animals [163]. Vaccinating fish may reduce the use of antibiotics in aquaculture. Duff [164] was the first to examine oral immunization against furunculosis in Atlantic salmon (*Salmo salar*). The advantage of using vaccines over antibiotics is that a vaccine stimulates the immune response and induces immunological memory, thus preventing future outbreaks by exposure to pathogens [165].

**Table 2.** Bacterial vaccines administered for tilapia.

| S. No. | Pathogens | Type | Mode of Administration | Efficacy | Performance | References |
|---|---|---|---|---|---|---|
| 1. | *Streptococcus iniae* improves the simulation of GALT (Gut-Associated Lymphoid Tissue) and specific antibodies | Attenuated | Intraperitoneal | 79–100% | Leads to higher antibody production conferred by cell-mediated immunity | [166] |
| | | | Bath | 86% | Leads to higher antibody production | |
| | | Formalin-Inactivated | Intraperitoneal | 79–100% | Provides good immunogenicity | |
| | | DNA Vaccine Modified PCI-neo plasmid or PBS (Streptococcal $\alpha$-enolase gene in pCI-neo plasmid) | Intramuscular | 72.5% | Leads to increased levels of proinflammatory cytokines and *S. iniae*-specific neutralizing antibodies | [167] |
| 2. | *Streptococcus agalactiae* | DNA Vaccine (Recombinant bacteria with surface immunogenic protein) | Oral | 75% | Immunogenic | [168] |
| | | Attenuated with erythromycin. | Intraperitoneal | 82–100% | Leads to higher antibody production | [169] |
| 3. | *Aeromonas hydrophila* | Heat-Inactivated Formalin Inactivated | Intramuscular | 90%, 86.6% | Immunogenic and facilitates highest antibody production | [170] |
| 4. | *Flavobacterium columnare* | Attenuated (Rifampicin-resistant low-virulence strains) subunit vaccine | Bath | 80% | Provides good immunogenicity and cross-protection to multiple genomovar co-infections | [171] |
| 5. | *Vibrio anguillarum* | DNA Vaccine (Recombinant flagellin A protein) | Intraperitoneal | Higher survival rate | Facilitates greater agglutination and bactericidal activity | [172] |
| 6. | *Edwardsiella tarda* | Whole-cell formalin-inactivated + recombinant GAPDH proteins that were emulsified with Montanide adjuvant | Intraperitoneal | 71.4% | Promotes greater antibody response | [172] |

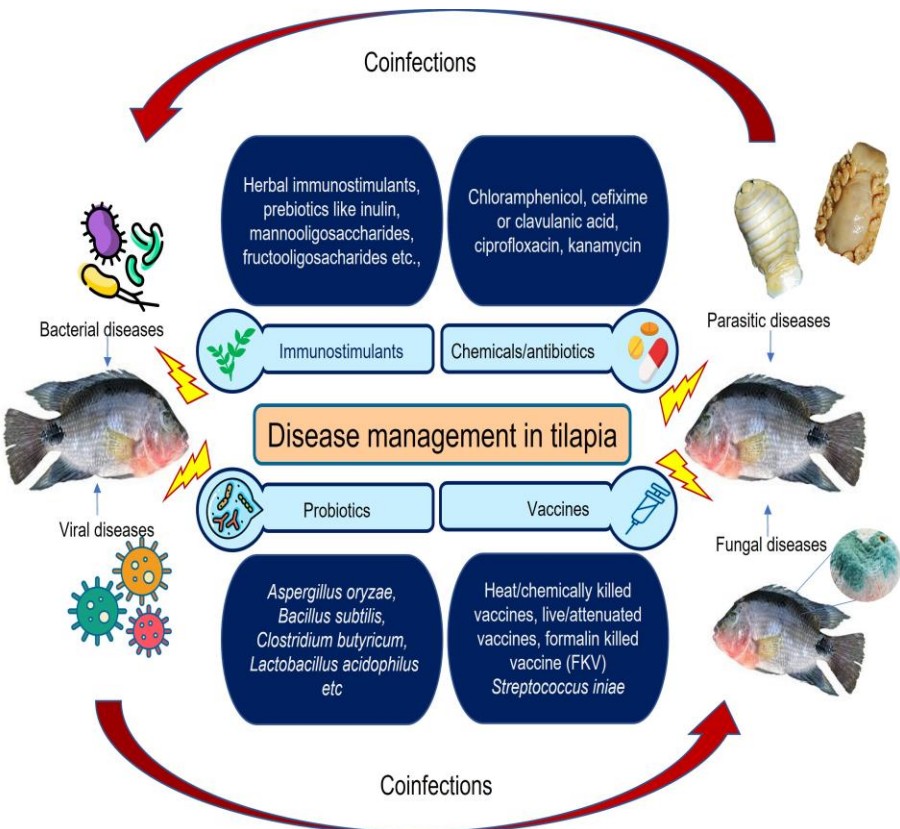

**Figure 6.** Steps involved in disease management in tilapia culture.

Attenuated and inactivated DNA- and RNA-type vaccines have been widely used to treat various bacterial, viral, and parasitic diseases in fish and have been experimentally tested in tilapia species. These vaccines can be monovalent, bivalent, or polyvalent [173–179]. Zhang et al. [180] suggested that understanding the mechanism of fish vaccination leads to higher defensive efficiency towards pathogens. The oral injection of an engineered formalin killed vaccine (FKV) for *Streptococcus iniae* administered to a red tilapia hybrid led to positive responses [181]. El tantawy and Ayoub. [182] reported that the inclusion of turmeric in fish feed combined with whole dead *A. hydrophila* cells led to a 100% survival rate in a group of *A. hydrophila*-infected tilapia. Table 2. shows that polyvalent vaccines consisting of formalin-inactivated *Streptococcus agalactiae, Streptococcus iniae, Enterococcus fecalis, Francisella orientalis*, and *Lactococcus garvieae* combined with the commercial adjuvant Montanide significantly increase the survival rates and immunogenicity of Nile tilapia [183,184]. However, commercial vaccines are not available in India [185].

*6.2. Antibiotics*

Commercial antibiotics are widely used to treat various fish diseases. Raj et al. [186] stated that *Aeromonas veronii* samples from diseased Nile tilapia exhibiting bilateral exophthalmia were sensitive to the following antibiotics: chloramphenicol, cefixime or clavulanic acid, ciprofloxacin, and kanamycin. However, the misuse or overuse of antibiotics impacts overall fish health and causes multidrug resistance in the pathogen [187]. There are also certain health concerns concerning the usage of antibiotics in aquaculture. For example, the gut microbiome of tilapia should not be altered as it promotes the growth and health status of the fish [188]. In this regard, effective technology has been developed to reduce the unwanted impacts of antibiotics by using absorbent material that delivers the antibiotics efficiently [189] (Table 3).

**Table 3.** Usage of antibiotics in tilapia culture.

| S. No | Antibiotic | Target Disease/Causative Organisms | References |
|-------|-----------|-----------------------------------|------------|
| 1 | Oxytetracycline | Francisellosis, motile Aeromonas septicemia, and Streptococcosis | [190] |
| 2 | Florfenicol | *Aeromonas salmonicida, Aeromonas hydrophila, Flavobacterium psychrophilum, Yersinia ruckeri,* and *Vibrio anguillarum* | [191] |
| 3 | Azithromycin | *Aeromonas* spp., *Pseudomonas fluorescens, Vibrio anguillarum, Flavobacterium columnare, Edwardsiella tarda, Streptococcus* spp., and *Enterococcus* spp. | [192] |
| 4 | Sulfamethoxazole | Alphaproteobacteria, cyanobacteria, Fusobacteria, and unclassified–P-proteobacteria | [193] |
| 5 | Erythromycin | Streptococcosis | [194] |

Módenes et al. [195] designed a mathematical modelling system for tilapia and tetracycline using an absorbent material (biochar) capable of absorbing this antibiotic and serving as a potential delivery method. It was shown that the use of a combination of natural compounds and antibiotics could be a method for reducing antibiotic resistance and other adverse effects. This study confirmed that rutin obtained from Citrus sinensis, a flavonoid compound rich in Vitamin P, combined with gentamicin exhibited better antibacterial and anti-biofilm effects against *Pseudomonas aeruginosa* [196]. The study also reported that rutin and the antibiotic florfenicol possess potential antibacterial and anti-biofilm properties both in vitro and in vivo against *Aeromonas hydrophila* [197].

*6.3. Immunostimulants*

Herbal plants are promising agents as they stimulate fish immunity at low doses without any side effects [198,199]. Their potential immunostimulants have significant natural characteristics, such as possessing low molecular weight, being water-soluble and amphoteric, and containing nitrogen molecules [200]. Immunostimulants in the form of chemicals, drugs, and natural compounds from plants and other sources can activate the host defense mechanisms against various disease-causing pathogens (Table 4). Bricknell and Dalmo [201] reported that immunostimulants boost the immune system of fish during larval development. Meena et al. [202] reported that beta-glucan can be used as a potential immunostimulant in aquaculture as it enhances the immune performance of fish. Beta-glucan and other immunosaccharides such as inulin, mannooligosaccharide, and fructooligosaccharide are widely used immunostimulants and are considered prebiotics. Immunostimulants, or immunopotentiators, improve the adaptive and innate immune system of the host [203]. Immunostimulants also serve as eco-friendly feed additions that can enhance a fish's growth and immune performance.

**Table 4.** Usage of herbal immunostimulants in tilapia culture.

| S. No | Immunostimulant | Organism | Performance | References |
|-------|-----------------|----------|-------------|------------|
| 1 | Turmeric (*Curcuma longa*) | Nile tilapia (*Oreochromis niloticus*) | Enhances growth, immunity, and antioxidant status | [204] |
| 2 | Pumpkin seed meal (*Cucurbita mixta*) | Mossambique tilapia (*Oreochromis mossambicus*) | Enhances growth, immune, and disease resistance activity | [205] |
| 3 | Velvet bean (*Mucuna pruriens*) | Mossambique tilapia (*Oreochromis mossambicus*) | Enhances innate immunity and growth performance | [206] |
| 4 | Ashwagandha (*Withania somnifera*) | Nile tilapia (*Oreochromis niloticus*) | Provides an immuno-therapeutic effect | [207] |

**Table 4.** *Cont.*

| S. No | Immunostimulant | Organism | Performance | References |
|---|---|---|---|---|
| 5 | Mangrove (*Excoecaria agallocha*) | Red hybrid tilapia (*Oreochromis niloticus*) | Enhances non-specific immune responses and disease resistance | [208] |
| 6 | Guava (*Psidium guajava*) | Nile tilapia (*Oreochromis niloticus*) | Enhances growth, nutrient utilization, and immune system | [209] |
| 7 | African wormwood (*Artemisia afra*) | Mossambique tilapia (*Oreochromis mossambicus*) | Enhances growth and disease resistance | [210] |
| 8 | Chamomile (*Matricaria chamomilla*) | Nile tilapia (*Oreochromis niloticus*) | Enhances growth and immune parameters | [211] |
| 9 | Spanish dagger (*Yucca schidigera*) | Nile tilapia (*Oreochromis niloticus*) | Enhances growth, hematology, nonspecific immune responses, and disease resistance | [212] |
| 10 | Oregano (*Origanum vulgare*) | Red belly tilapia (*Coptodon zillii*) | Enhances innate immunity | [213] |
| 11 | Peppermint (*Mentha piperita*) | Nile tilapia (*Oreochromis niloticus*) | Enhances hemato-immunological parameters | [214] |

Bustamam et al. [215] reported that a 2.5% inclusion of *Isochrysis galbana* (IG) supplemented as a dietary immunostimulant enhances the immune system of red hybrid tilapia. It also increases the abundance of certain secondary metabolites such as glutamate, isoleucine, and tyrosine. Notably, immunostimulants tend to alter the metabolomics of the fish, which alters their metabolism [216].

*6.4. Probiotics*

Live microorganisms that can improve host health are collectively referred to as probiotics. The common probiotics used in aquaculture include the *Aeromonas, Bacillus, Clostridium, Cornybacterium, Enterococcus, Enterobacter, Lactobacillus, Lactococcus, Pseudomonas, Shewanella, Saccharomyces,* and *Vibrio* species [122]. These potential probiotics tend to enhance the growth and immune system of fish [217]. Essa et al. [218] reported that tilapia growth performance and the activity of digestive enzymes such as amylase, protease, and lipase were improved by providing *Bacillus subtilis* and *Lactobacillus plantarum* or a mixture of yeast (*Saccharomyces cerevisiae*) as an alternative feed. Moreover, these probiotics were associated with the gut microbiota and enhanced the enzymes that hydrolyze macronutrients for the better digestion and absorption of nutrients [115]. Ghosh et al. [219] investigated the probiotic and antipathogenic nature of *Bacillus sp.* Banerjee and Ray. [220] experimented with the antagonistic effects of *Bacillus megatarium* in the intestine of tilapia. Certain species of *Bacillus* can degrade cellulose. *Bacillus circulans* isolated from the gut of tilapia increased the fermentation of cellulose [221]. Lara-Flores et al. [222] stated that probiotics incorporated in a diet consisting of 40% or 27% crude protein improved feed conversion ratios and weight gain compared to a control diet. Probiotics not only promote growth but also improve the immune system, disease resistance, and survival rate of tilapia. Aly et al. [223] fed sample fish a mixture of *Bacillus subtilis* and *Lactobacillus acidophilus* as a probiotic, which resulted in a significantly higher survival rate in Nile tilapia. Samat et al. [224] attempted the administration of a probiotic via live feed. *Moina micrura* was used as the live feed and *Bacillus pocheonensis* as the probiotic. This combination resulted in the improved health and survival of the fish. Ringo et al. [225] reported that *Bacillus amyloliquefaciens* supplemented as a probiotic in feed for tilapia modifies the gut microbiome and enriches the production of secondary metabolites. The major criteria for the supplementation of probiotics to fish vary based on the species and depend on the concentration, mode of administration, etc. [122].

## 7. Projects Developed for The Production of Tilapia in India

Governing bodies such as the National Fisheries Development Board (NFDB) and the Rajiv Gandhi Center for Aquaculture (RGCA) have given sustained and focused priority to the fisheries sector through policies and financial support designed to support small-scale farmers, women, and various centers in order to achieve sustainable fish production in India (Table 5). The RGCA, in association with the World Fish Centre (WFC), developed genetically improved varieties of tilapia for the betterment of fish farmers and small householders, thereby helping to promote tilapia farming and improve local economies in the country. The WFC also focuses on sustainable and logical breeding programs for the tilapia industry in India [28,226].

**Table 5.** Projects and schemes for tilapia culture in India.

| S. No. | Governing Body/Funding Agencies | Project | Target Fish Species |
|---|---|---|---|
| 1. | NFDB | Brackish water cage culture for sustainable aquaculture in coastal regions of India | Milk Fish (*Chanos chanos*), Asian seabass (*Lates calcarifer*), grey mullet (*Mugil cephalus*), pearlspot (*Etroplus suratensis*), Nile tilapia (*Oreochromis niloticus*), silver pompano (*Trachinotus blochii*) |
| | | Demonstration of azolla production for tilapia feed supplement in Madhavaram, TNJFU Campus, Tamil Nadu | GIFT Tilapia |
| | | Backyard Recirculatory Aquaculture System | Monosex tilapia, *Pangasius valenciennes* |
| 2. | RGCA working in association with (WFC) to enhance the genetic strains of tilapia. | Establishment of a satellite nucleus of the GIFT strain at RGCA to support tilapia production in India: Phase I (2011–2016) Establishment of a satellite nucleus of the GIFT strain at RGCA, India: Phase II (2019–2023) | GIFT Tilapia |

The Indian government's policies and goals for the fisheries sector have been strengthened by FAO activities. The Bay of Bengal Program (BOBP), a regional fisheries program created by FAO, is centered in Chennai, India [13]. Through collaboration with global aquaculture and fisheries allies, India is contributing to the share of global public goods, including by sharing its expertise in agriculture (aquaculture) and rural development with other developing countries. In 2022, the Indian government launched Pradhan Mantri Matsya Sampada Yojana (PMMSY) to form a blue revolution by enhancing the sustainable development of fisheries and aquaculture (Figure 7). This program creates various employment opportunities. In addition, this program is collaborating with various private organizations such as Fountainhead Agro Farms Private Limited to enrich the production of tilapia using Israeli technology.

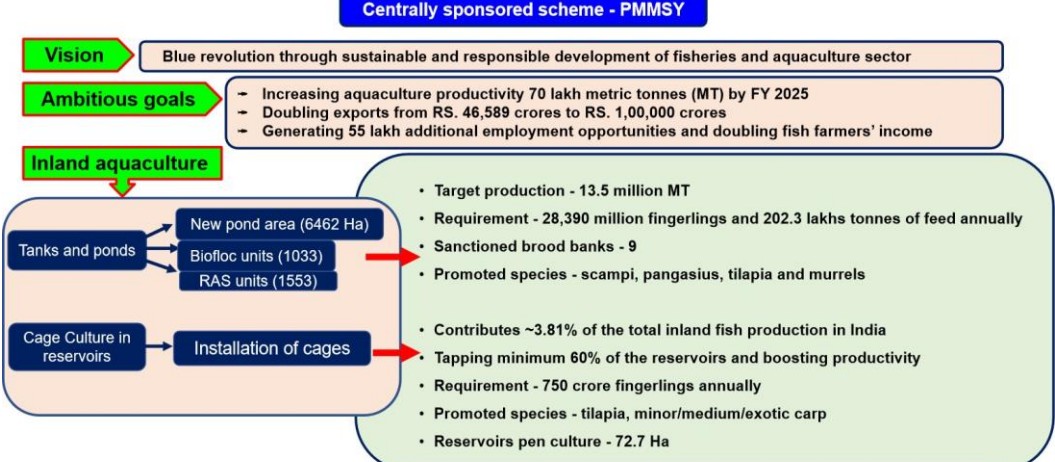

Source: Inland fisheries in India, Department of Fisheries, Government of India
(https://dof.gov.in/inland-fisheries)

**Figure 7.** The schematic representation of aims and expected outcomes of centrally sponsored scheme (PMMSY) with the potential to increase aquacultural productivity, exportation, and employment opportunities.

### 8. Blue Economy—Future Perspectives

The blue economy is critical to tilapia production and is predicted to grow dramatically in the upcoming years. The tilapia sector is under pressure to enhance its productivity while reducing environmental concerns as the demand for food increases [227]. With the global population estimated to exceed 9 billion by 2050, the tilapia sector will be critical in fulfilling the increasing need for protein. The future of tilapia production is bright because of several elements that support the rise of the blue economy [228]. This review has highlighted the critical factors defining the tilapia industry's future. To begin with, technological advances in tilapia farming have revolutionized the sector. Conventional agricultural practices are no longer appropriate for today's commercial market needs. The Blue Economy has created new prospects for international commerce, which has led to greater growth in the tilapia sector. Foreign investment is being driven by rising global consumer demand, and trade agreements are simplifying market access for many nations. Furthermore, the increasing demand for live fish, such as tilapia, gives providers additional potential to develop the market beyond the commonly sold frozen fish. In summary, the future of tilapia production from the perspective of the blue economy seems promising. Technological advancements, advances in fish feed production, shifting consumer habits, and possibilities regarding international commerce are all contributing to this expansion. The industry's sustained growth should help to drive economic development and food security by fulfilling the rising consumer demand for healthy, sustainable foods.

### 9. Conclusions

In India, aquaculture is a promising economic activity and a rising sector with wide resources and potential. The vibrancy of the aquaculture sector could be visualized as a drastic advancement in the field of aquaculture, which India has achieved in past decades. Tilapia significantly contributes to the total share of aquaculture exports in India, which boosts the country's economy. With recent breakthroughs in aquacultural technology and improvements in the diets of tilapia, there has been constant advancement in tilapia output, leading to the sustainable development of Indian aquaculture. Tilapia cultivation may be an economically feasible choice for aquaculture production in various locations of India, as long as suitable investment and management practices are employed. The Rajiv Gandhi Centre for Aquaculture (MPEDA, Ministry of Commerce and Industry, Government of India) has established a tilapia project and breeding program focused on the use of potential GIFT strains to improve production conditions in India in collaboration with the WFC,

Malaysia. The NFDB of India was formed in 2006 and is an autonomous body under the Ministry of Fisheries, Animal Husbandry, and Dairying of the Government of India that seeks to promote and encourage tilapia farming. Still, farmers are facing difficulties related to disease management while culturing tilapia, necessitating the provision of vaccines for longer-term protection and low-cost vaccines that increase mucosal immunity. Various technologies and tools are available that can support the future of aquacultural production and the betterment of the country's economy and food supply. The policy making regarding tilapia aquaculture in India not only aspires to promote economic value but also concerns ensuring national and global food security, diminishing malnutrition, and reducing poverty.

**Author Contributions:** All authors contributed to this study's inception and design. M.A. and S.J. wrote the first draft of the manuscript, which was reviewed and edited by A.S., V.V., P.B.B., Z.A.K. and G.T.-I. Finally, T.R. reviewed, edited, and supervised the work. All authors have read and agreed to the published version of the manuscript.

**Funding:** Research funding was supported in part by funds provided by USDA-NIFA Sustainable Agriculture Systems, Grant No. 2019-69012-29905. Title of the project: Empowering US Broiler Production for Transformation and Sustainability USDA-NIFA (Sustainable Agriculture Systems): No. 2019-69012-29905.

**Data Availability Statement:** Data will be made available on reasonable request.

**Acknowledgments:** The first author is grateful to Bharathidasan University for providing University Research Fellowship (Ref. No. 026525/URF/DIR-RES/2020; dt: 04.01.2020). The authors (M.A., S.J., A.S., V.V. and T.R.) are thankful for UGC-SAP-DRS-II (F.3-9/2013(SAP-II), the Department of Science and Technology-Fund for the Improvement of Science and Technology Infrastructure (DST-FIST) Level-I (stage-II) (Ref. No. SR/FST/LSI-647/2015(C); Date.11.08.2016) and the Department of Science and Technology for the Promotion of University Research and Scientific Excellence (DST PURSE Phase - II) (Ref. No. SR/PURSE PHASE 2/16(G)/and 16(C); Date. 21.02.2017) of the Department of Animal Science, Bharathidasan University, for allowing our use of their instrumentation facility. The authors also thank "RUSA, 2.0-Biological Sciences, Bharathidasan University". The authors greatly acknowledge Paul B. Brown for critically reviewing the manuscript.

**Conflicts of Interest:** The authors declare they have no conflict of interest.

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
