# Peer review of "Recent Advances in Tilapia Production for Sustainable Developments in Indian Aquaculture and Its Economic Benefits"

_fishes, doi:10.3390/fishes8040176_

Round 1

Reviewer 1 Report

The manuscript is a report on the state of art and the challenges of tilapia aquaculture in India reporting on the main farming schemes, challanges of the different technologies, funds available and projects.

It is well structured and clear and provide a number of information hence largely contributing to provide a clear picture of the tilapia production in India.

Nevertheless, it appears to be more a report than a review. What is missing, in my opinion (and in line with the journal guidelines about the Review’s content) is a Discussion section highlighting any gaps in the research or management concernign tilapia production in India and indicating what the “future direction” of the sector could/should be from a Blue economy perspective.

The title include the expression “its economic benefit”. Nvertheless, economic figures about production and growth are included only in the introduction and section 2. Also section 3 should talks about contribution of India to Tilapia production but it’s unclear, from the reading, if it just talks about figures about Indian fisheries. I would suggest a table summarising the role of Indian fisheries on the global production (even if in % terms) as well specific figures on the Tilapia sector (volume, value, number of farms, employment created). If possible, also highlight the impact (increase?) of the Governmental programmes/projects.

The text contains a lot of acronyms (due to the technicity of some sections). Please, be sure that all the acronyms are explained at the first time and are reported after the extended text (in parenthesis). I found one not explained, GIFT (Genetically Improved Farmed Tilapia). Some acronyms are explained more than once (e.g. WFC at line 529).

Specific comments: 

Line: 372 “Lack of any of the essential nutrients can cause malnutrition, compromise the immune system which in turn resulted in poor growth”. Redundant, it’s a common sense conclusion

Line 377: “Diets must provide fish carbohydrates, proteins, fat, essential fatty acids, amino acids, vitamins, and minerals”. Redundant, it’s a common sense conclusion

Line 379-381: editing needed. There are few corrections along the overal manuscript to be made, e.g. comma missing, verb tens inappropriate, please check.

Table 2: change formatting, now difficult to read (maybe horizontal would be better)

Figure 6: redundant, it does not add anything to the reading

Table 3: move table inside text of section 7.2

Table 4: move at the end of section 7.3 (do not start section with tables)

Line 540: use capital letter for enterprises names (i.e. fountainhead agro farms private limited)

Line 548-543: text not appropriate for a conclusion section.

Author Response

Please see the attachment for Author's response to reviewer 1.

Reviewer 2 Report

Review

A comprehensive and documented review, thoroughly reporting on the state of art and the challenges of tilapia aquaculture in India.

The authors use a clear and concise language.

A large pool of bibliographic resources (almost 200) indicates the great documentation effort.

I recommend the publication of the manuscript after a minor language and spelling revision and addressing the specific comments below:

Specific comments

Lines 5 and 6: The last letters of authors’ names are probably supposed to be superscripts indicating the affiliation. Please correct.

Line 18 (Abstract): Revise the phrase by correcting word order “contributes to and promotes aquaculture practices“.

Lines 20-21 (Abstract): “Indian aquaculture currently stands second in aquaculture 20 production“. Worldwide? Please be specific in the abstract.

Line 39: The sentence is incomplete, revise it.

Line 49: Do you you mean “Tilapia can tolerate osmotic and alkalinity stress?“. Correct accordingly.

Line 60: Jaguar guapote is the common name, no italics are needed. Provide the scientific name as well.

Line 66: Replace “are“ with “is“.

Line 67: A preposition is missing after 7.56% (to?).

Line 96: No full-stop neede in Tor putitora.

Line 100-104-116: Decide whether you write the name of the species in capital letter or not and be consistent throughout the entire manuscript.

Line 148: An article is required “A Steering Committee...“.

Line 170: No full-stop needed after “Oparinde“.

Line 180: The established term for RAS is “Reciculating Aquaculture System“. Please use it everywhere in the manuscript.

Lines 197-199:  A predicate is missing. Revise.

Line 261: Avoid using the plural of “zooplankton“.

Line 284: Similarly, avoid using the plural of “phytoplankton“.

Line 293: Put the abbreviation (IPUAS) after the full phrase.

Line 302: Scheme: again, avoid the plural of “phytoplankton“ and “ zooplankton“.

Line 303-304: Make the grammatical agreement between the plural subject and the predicate.

Line 345: Remove the underscore at the end.

Line 356: Remove the underscore after “tilapia“.

Line 361: Agree the predicate with the plural subject.

Line 362: The predicate is missing after Avallone et al. [84]. Rephrase.

Line 371: Agree the predicate with the plural subject.

Line 377: Remove the underscore before “proteins“.

Lines 379-381: The sentence is unclear. Rephrase in a more indicative manner.

Line 382 (Table 1): Be consistent with the use of the full-stop in the cells of Table 1. Some cells have a full-stop, some don’t. Moreover, be consistent with the italization of all scientific names of plants, bacteria etc. in column one. Common names should not be italized.

Line 388: Remove the underscore before “Fish“.

Line 408: Capitalize “Flavobacterium“ and remove italics from “and“.

Line 411: Capitalized the names of the monogenean genera.

Line 416: A verb is missing and the sentence is unclear. Rephrase.

Line 432: The sentence is unclear. Rephrase.

Line 437: Reduce the font of Table 2 text, so to avoid word separation in the cells. Rearrange the table.

Line 440: Remove “of“.

Line 474: A full-stop is needed after “culture“.

Page 17: The header of Table 4 is misplaced.

Line 488: Remove the comma after the reference number.

Line 497: “and“ and “species“ should not be italized.

Line 519: Remove the comma before the reference number.

Line 1087: The Disclaimer should not be included in the reference numbering sequence.

Author Response

Please see the attachment for Author's response to reviewer 2.

Reviewer 3 Report

The review offers a nice overview of the tilapia aquaculture in India, it is quite big with a lot of references, sometimes facts are listed after another without real red thread, It would be better to perhaps be more synthetic on some topics, select/summarize the (relevant) information, use better transitions within and between the different sections and subsections.

The different subsections should be introduced in the introduction, some parts should be regrouped under a higher subsection level: i.e. there is no section 5, the authors start with ‘5.1’; sections 5.1-5.5 are dealing with the different culture methods, thy should be grouped under a section 5 ‘culture methods’, sections 5.6-5-8 deal with species/sex selections’, they should be under a new section (6), then  section 6 should be 7, and section 7 should be 8

The abstract, and later the introduction (which is a big block), display a lot of information but not real connection/transition between the different facts listed. The authors also need to clearly state the goals of this review at the end of the introduction (it is nicely done in the abstract) and introduce the different parts of the study (i.e we will review the current situation of Tilapia aquaculture in India and current guidelines, also give an overview of the different culture systems, the species selection, feed and disease managements.

Specific comments
Authors affiliation, L9 and L11, please be consistent, either use the stare abbreviation or not

L12, no need to use capitals for the name of the corresponding author

Abstract

L18 no ‘s’ at ‘to promote’

L21 (and also elsewhere in the manuscript), the authors introduce abbreviation (i.e SDG L15), but don’t use them afterwards (i.e L21, ‘sustainable development goals), or repeat the abbreviation in other part of the manuscript. It is the same with the Latin name of the fish species, the English name is often followed by the Latin name, it is  overcrowding the manuscript.

Keywords

I don’t think SDG is a very relevant keyword, fisheries as well, resources is very vague, food and nutrition security is very long, maybe you could be more specific regarding ‘aquaculture’, replace it by ‘India aquaculture’, or ‘tilapia aquaculture’ which are the themes of your review. You should also mention systems since you present different types of systems for this species, species selection as well, disease management

Graphical abstract. Good choice of illustration, good keywords representing your study, but the arrows used around the fish may be a bit misleading. The authors use capital letters sometimes, other times not (i.e. Carbon Nitrogen rich feed; Microbial protein; Disease Management Strategies…) please be consistent, I would discourage the use of capitals in a sentence

Introduction,
L36, replace food insecurity by food security?

L36 and L41, be consistent, SDG or SDGs

L45-46 and 46-47 these statements need a reference
L50-51, you need to add a reference in number [6], not ‘World bank (2013)’ to be consistent with the journal’s reference citation

L58, there is a ‘(‘ missing before  Oreochromis niloticus

L60 you start to talk about hybrids, then you switch to history of tilapia aquaculture, and you go back to hybrids L66. It is a general issue with the introduction, there is not red thread or connection between the different parts, but a lot of facts stated one after the other, without real order.

L60 ‘Jaguar guapote’ should not be in Italic, it is the English name of the species, you should add Parachromis managuensis, the Latin name, after it

L67-68. Those data are a bit old, the authors should look at the FAO 2020 or 2022 data, or the program FishStats from FAO to use the latest data

L71-80, this part should be reformulated, the authors talk about the Neel Kranti program stating about its goals to increase FISHERIES production for several lines, and the L79 the authors mention fisheries and aquaculture, the goals of this program, and especially more towards aquaculture than fisheries should be clearly presented,

L80, please add the Latin name of Indian carp

L89, here you should introduce the different parts of your review

Part 2-3-4; maybe they could be regrouped under a subsection 2, and be 2.1; 2.2 and 2.3, rather than parts 2,3 and 4,  here you talk about the situation and protocols/projects/figures on India production on tilapia

L91 and elsewhere in the manuscript. You have already mentioned Oreochromis niloticus L58, no need to mention both English and Latin name of a species after the first time

L96, no ‘.’ After ‘tor’

L99, 100, 117, 132, 134 use number to quote a study, not ‘(NFDP 2015 and Johnson et al 2020 and 2022)’, ‘FAO 2020’, MoFAHD 2020

L100, not capital letter for tilapia

L105, no ‘.’ After ‘Panikkar’

L117, use FAO 2022, more recent, also you already abbreviated FAO L117, no need to do it L132 again, use ‘FAO’ from now on

L123, you mention Indian aquaculture is doing great, but you talk about its rank within capture fisheries, please use aquaculture data, L123-129, you focus on fisheries, it should be about aquaculture

L137, use ‘high density’ rather than ‘overpopulation’

L160 and elsewhere, ‘biofloc’ instead of ‘bio floc’ (no space)

L168, you need a reference to back up this statement

Sections 5.1-5.5; you need to have a bigger title for the ‘Part 5’ (missing), L179-180, that could be ‘production systems’?

Figure 1, remove ‘Recirculatory aquaculture system (RAS) from the figure, it is already in the legend, ‘biological filter’ instead of bio filter system’ Disinfection: add ‘UV’? Change oxygen to O2, since you use ‘CO2’. Please check the direction of the arrows, they should go from the mechanical filter to the biofilter?

L183-193, you need to explain a  bit more the functioning of RAS system, i.e. you only mention the removal of ammonium but you don’t mention how (i.e by nitrifying bacteria transforming ammonium in nitrate, via nitrite), and the necessity to either have regular water exchanges to limit accumulation of nitrate, or the need to have anaerobic denitrifying filters (not present in your figure 1), you should say your figure 1 is a partial RAS with just nitrification, it should be stated in the legend/text.

Delete the double spaces L185, 278, 291, 386, 449, 515,

Figure 2, same as Fig 1, you can remove ‘BIOFLOC TECHNOLOGY’ from the figure, it is in the legend.
Also, please harmonize the terminology, in the figure you write biofloc, in the legend Bio-Floc, and in the text Bio Floc, please use ‘biofloc’

L197, you already introduced the abbreviation L160, you should use BFT

L199-200, replace Hargreaves 2006 by a number

L213 ‘optimum production’ rate is quite vague, you should be more specific

L226, please rephrase, what do you mean with ‘Supplementary feed’?

L231, ‘Optimal stocking density’, this sentence is vague, You should give figures about it, what are those optimal densities (give a range and reference). You say L236 50 fish mono-sex Nile tilapia/m3, but you don’t specify the weight/age of the fish, it is better to give densities as kg/m3, rather than individuals/m3

L239, please rephrase (then provision of natural feed lessens)

L245-246, what do you mean by toxic gases? The main issue is the release of nitrogen nutrients, no real gases in cage farming

Sections 5.4-5.5 and 5.5.1 should be rearranged a bit. You have a 5.5.1 but no 5.5.2, it is not necessary to make a subsection. Moreover, this section about IMTA should go together with the 5.4, polyculture, since in the polyculture you are talking abut aquatic organisms, and in the 5.5 you talk about integration of tilapia culture with terrestrial livestock.

L254-255, you should add figures; how many days/months are we talking about when referring to the growing period.

L259 (x2), L263, you should add the Latin name of those animal species, same L280, you should add the Latin name here, the first time you mention this species, not L296. Also, add a capital.  

Part 5.5, the integrated farming. Here you should discuss about the potential health and safety issues using wastes (resources) such as feces from livestock. L274-279, you introduce this section as integrating fish with terrestrial animals, but L279-283, you switch to fish and prawn, it is not very logical, this part is better fitting with the IMTA

L285, as mentioned earlier, you should rearrange the sections 5.4-5.5, and change 5.5.1 since there is no 5.5.2

Figure 3, you have two times seaweeds in the figure, on the top left and top right, just use one.
Change ‘recycler’ to ‘detritivore’

Parts 5-6-5.8. Those should be grouped under a separated section than the previous one since it is another theme (i.e selection/genetic)

L315-315, please add a reference to back up this statement

L317, this sentence is a bit out of the blue, maybe develop a bit more?

L318 ‘The study indicated’, which study (also delete the ‘_’)

L321, ‘due to some concerns’, please develop more

Figure 4, delete the title in the figure, you have it already in the legend.

You should add the % of each generation (i.e for the F1 YY males are 100%), XX natural female. c.a 50%, since you mention that you have 100% of XY in the F2

L329, use GIFT straight away, you have already described the abbreviation earlier

L332 ‘current strains’, it is too vague, please specify

L336, here you can use directly FCR, you already explained the abbreviation L268

L343, please delete ‘_’

L351-355, it is a long list of species, not very informative as well, delete? Or if you really want to keep it make a table in the supplementary material and refer to it (+add the % for each crossing?)

L356, L388 please delete ‘_’

Section 6

You should change this part, you are talking about the importance of fishmeal, but it can be misleading for tilapia, they are omnivorous (mostly when young), once adult, they tend to be more herbivorous. The diet preference/depending on the stage should also be mentioned. You should focus on the need for tilapia/need for tailor made feed rather than talk about fishmeal replacement. L380, since they are herbivorous, anti nutritional factors may not as a big deal as for carnivorous species like salmonids. The main issue that should be also discussed is that to make more feed for tilapia would be the use of arable land to make fish feed, while these could be better used for human food directly. L380, no need to use capital letters for anti-Nutritional

L375, use directly FCR, already explained L268

Table 1,

Change the order of the references! Make it in the numerical order, the last reference in the text is 89 (L381). The first line of the table, the reference is 150, it should be 90, the you go to 95, 96 ect, it should be 91, 92…

Silybum marianum, in italic, Oreochromis niloticus should be between brackets

Salvadora persica, in italic, write the full genus name for ‘A. Hydrophilia

Methol essential oil, Dietary coenzyme ect… (refs 100-106, ref 111 to 115) when not a Latin name: don’t use italic. ‘organic’, use capital O, don’t use the abbreviation WPC, if you don’t need it later,

Ref 102, 93, 103 and 109, you use ‘O.’ for the second species of the crossing, don’t do that, use the full name to harmonize.

Ref 106 ‘and’ should not be in italic, Bacilus pumilus should be in italic, ‘and exogenous protease’ should not be in italic

Ref 108 and 109, the Latin names should be in italic

Part 7,

L405, the species quoted should be in italic (Latin names)

L408, ‘and’ not in italic

L 416-418, why do you use ‘’ ‘’ for Streptococcosis? No need, to harmonize with the rest of the text.

L422, here you can introduce the necessity to use vaccination and better hygiene protocols to avoid the use of antibiotics (sine you are going to speak about vaccine later), it is a good transition sentence.

L 426, L533 please delete ‘_’

Figure 5, ‘B’ of bacillus should be in italic, the fish representing fungal disease is not a a tilapia, it’s a scalare (angelfish), please change

L432, Atlantic salmon + add the Latin name as well

Table 2, please check the format, some of the text is shifted, it doesn’t read very well (I,e ‘administrati’ in on one line then ‘on’ is later., same for ‘intramuscula’ ‘r’,  ect. Make the font smaller/maybe reduce the text in the second and fifth columns a bit?

L452, ‘commercial or synthetic antibiotics’, I don’t really understand the difference, perhaps rephrase? Synthetic can be commercial.

Figure 6. This figure doesn’t add anything to the review, please delete.

Table 3,

L474, no capital letter for ‘Herbal’.

I don’t think you need to abbreviate OCT, you don’t use it later

Same issue with the reference for here and the text,L463, you quote ref 158, then L464 you are at ref 164, and in the table, you start with ref 159, the table comes after the text, so change those number to match the numerical order/order of the text/table

Ref 160, write the full genus for ‘A. hydrophylia

Ref 160, harmonize ‘spp’ use ‘spp.’, in italic, and with a dot, for both, same for L504

Table 4 the legend seems to be glued to Table 3, make it separated/with the correct font + add a more detailed legend, the ‘performance’ is maybe a bit vague, you should say ‘improved’ or ‘reduced’.

Table 3 and 4, the first column you use ‘S.No’ or ‘No’, but not in tables 1 and 2, you should harmonize (and also harmonize the title of the column)

No 2, you need a space between ‘meal’ and ‘(Curcurbita’

No6, 8 and 10, remove the ‘L’, you don’t mention it for other species. No6, add a capital letter at ‘Grow’
L482-483, harmonize how you write beta glucan (use ‘-‘, or not)

L500, use the full genus name for ‘B.’
L522, 526 and table 5 (three times) RGCA abbreviation is already explained L113, use the abbreviation directly
L529 and Table 5, use WFC directly, you already abbreviated it L526

L531, you can also use FAO, already abbreviated earlier

Table 5, last column, first line, remove ‘and’; lines 3, 4 and 5, you should also mention some Latin names not mentioned earlier (i.e Pangasius)
L552, NFDB abbreviation was already described L552.

L553, remove ‘.’ And replace ‘Govt.’ by government

Ref, check ref 9, change the link to researchgate to doi? Also, see my previous comments, make sure that the few references ‘author-date’ in the text are changes to numbers and update the reference list, and also make sure the numerical order is respected in the text/tables.

Author Response

Please see the attachment for Author's response to reviewer 3.
